# Enhanced Crystallinity and Luminescence Characteristics of Hexagonal Boron Nitride Doped with Cerium Ions According to Tempering Temperatures

**DOI:** 10.3390/ma14010193

**Published:** 2021-01-03

**Authors:** Jae Yong Jung, Juna Kim, Yang Do Kim, Young-Kuk Kim, Hee-Ryoung Cha, Jung-Goo Lee, Chang Sik Son, Donghyun Hwang

**Affiliations:** 1Division of Materials Science and Engineering, Silla University, Busan 46958, Korea; eayoung21@naver.com; 2School of Materials Science and Engineering, Pusan National University, Busan 46241, Korea; kja6037@pusan.ac.kr (J.K.); yangdo@pusan.ac.kr (Y.D.K.); 3Powder & Ceramics Division, Korea Institute of Materials Science, Changwon 51508, Korea; voice21@kims.re.kr (Y.-K.K.); h.cha@kims.re.kr (H.-R.C.); jglee36@kims.re.kr (J.-G.L.)

**Keywords:** hexagonal boron nitride, photoluminescence, cerium, anti-counterfeiting, crystals

## Abstract

Hexagonal boron nitride was synthesized by pyrolysis using boric acid and melamine. At this time, to impart luminescence, rare earth cerium ions were added to synthesize hexagonal boron nitride nanophosphor particles exhibiting deep blue emission. To investigate the changes in crystallinity and luminescence according to the re-heating temperature, samples which had been subjected to pyrolysis at 900 °C were subjected to re-heating from 1100 °C to 1400 °C. Crystallinity and luminescence were enhanced according to changes in the reheating temperature. The synthesized cerium ion-doped hexagonal boron nitride nanoparticle phosphor was applied to the anti-counterfeiting field to prepare an ink that can only be identified under UV light.

## 1. Introduction

Hexagonal boron nitride (h-BN) is an important material with excellent properties, such as a wide band gap (4.4~6.0 eV), high electrical insulation, low dielectric constant, high temperature stability and large-scale oxidation cross-sectional area, and thermal neutrons [1,2,3]. All these features have made h-BN a promising material in the aviation industry. Moreover, it has many uses in microelectronic mechanical systems (MEMs), biomedicine, fireproofing, laser devices, solid-state neutron detectors, lubricants and electrical insulators [4,5,6]. Among the features of h-BN that make it suitable for such varied uses, the wide band gap allows it to be used in applications that require a material with unique luminescence characteristics.

Since the first observation of intense far-ultraviolet (UV) excitation emission, the unique optical and fluorescent properties of h-BN have attracted special attention in the past ten years, making this material a candidate for new light-emitting devices, and it is expected to be used as a future material for photovoltaic applications [7,8,9]. In addition, the band gap energy of h-BN can be adjusted by designing a super lattice, doping, recombination of organic functional groups and surface functionalization. Doping with rare earth (RE) ions is the most widely known method of tuning the band gap energy of h-BN. Continued interest in rare earth-doped nitride-based materials appears to be increasing, especially for h-BN, partly because of the discovery that the thermal quenching of luminescence decreases as the bandgap of the host materials increases [10,11]. In particular, metal nitrides or oxynitrides containing activator ions are often used as color-converting phosphors for white light-emitting diodes (wLEDs), because they have excellent visible light emission and less thermal quenching. However, most nitride-based host materials used in phosphors are ternary or quaternary nitride semiconductors. Binary nitride semiconductors, such as aluminum nitride (AlN), gallium nitride (GaN) and BN are rarely used as host materials for luminescent activators [12,13,14]. Steckl et al. proposed a diode that emits green light in the visible region by doping erbium ions using GaN as a host material [15]. In a study by Jadwisienczak et al., AlN thin film doped with terbium ions by sputtering was grown on a silicon wafer and a light-emitting device in the visible region was produced by a re-heating process [16]. Kim et al. investigated the changes in the luminescence of deep blue with various doping concentrations of cerium ions using h-BN as a host [17]. Mauro et al. described the optical properties of dilute semiconductor materials by presenting a set of highly efficient analytical equations that focus on the evolution of the peak luminescence gain with temperature and the relationship to sample quality [18].

In this study, the crystallinity and luminescence properties of cerium ion-doped hexagonal boron nitride (h-BN) according to the change of reheating temperature were investigated by X-ray diffraction, transmission electron microscopy, Raman spectroscopy, photoluminescence and photoluminescence excitation analysis.

## 2. Materials and Methods 

### 2.1. Synthesis of Hexagonal Boron Nitride Nanophosphor Doped with Ce^3+^

The experimental procedure is schematically shown in Figure 1. Hexagonal boron nitride nanophosphors were synthesized from boric acid (H_3_BO_3_, Sigma-Aldrich, St. Louis, MO, USA, ≥ 98.5%) and melamine (C_3_H_6_N_6_, Sigma-Aldrich, 99%). First, 7 mmol of H_3_BO_3_ and 1 mmol of C_3_H_6_N_6_ were dissolved in 150 mL deionized (D.I) water. Cerium 0.05 mmol was also incorporated by dissolution of cerium nitrate (Ce(NO_3_)_3_·6H_2_O, Sigma-Aldrich, St. Louis, MO, USA, 99.999%). These materials were completely dissolved in D.I water, evaporated at 120 °C while stirring at 500 rpm to obtain the precursor, and then dried at 80 °C for 24 h. The dried precursor underwent pyrolysis in an alumina tubular furnace (AJEON FURNACE, Namyangju, Korea) at 900 °C under nitrogen atmosphere. After heat treatment, the sample was rinsed with D.I water to remove any remaining unreacted material. When the powder was collected by agglomeration in the direction of gravity using a centrifuge, the solution was discarded, and the recovered powder specimen was dried at 80 °C for 12 h. The samples of h-BN synthesized at 900 °C were subjected to re-heating at 1100, 1200, 1300 and 1400 °C for 2 h under nitrogen atmosphere to investigate their crystallinity and luminescence in relation to the tempering temperature.

### 2.2. Characterization of h-BN Nanophosphors 

X-ray powder diffraction (XRD, Rigaku, Ultima IV, Tokyo, Japan) analysis was performed to investigate the crystallinity of the samples in relation to various temperatures with Cu-Kα radiation (0.15406 nm) generated at 40 kV and 20 mA. The measurement range was performed from 20 to 80 °C and the step scan was performed for 4 s at 0.02 degree intervals. Raman spectra were recorded using a dispersive Raman spectrometer (LabRam-HR 800, Horiba Jobin-Yvon, Longjumeau, France) equipped with a microscope and a 522-nm laser as the excitation source. In addition, the specimen was made into a 1 mm × 1 mm cylindrical pellet and measured in the range of 700 to 2300 after focusing at 1800 gv magnification and focus using the E2g mode of the optical lens. The photoluminescence (PL) of the samples was measured and photoluminescence excitation spectroscopy (PLE) was conducted with a fluorescence spectrometer (FP-6500, JASCO, Tokyo, Japan) equipped with a xenon (Xe) flash lamp. At this time, the supplied energy of the spectral lamp was measured with 200 photomultiplier tube modules (PMT). The PL measurement range was from 250 to 800 nm, and the PLE was from 200 to 550 nm. The morphology and crystallinity of the nanophosphors were observed by transmission electron microscopy (TEM, JEM 2100F, JEOL, Tokyo, Japan). The element distribution was analyzed by energy dispersive X-ray spectroscopy (EDX, X-Max 150, Oxford Instruments, Abingdon, UK) to confirm the presence of rare earths in the samples. The resolution was about 129 eV and the analysis time was X-ray exposure for 2 min for each specimen. At this time, the components were compared with B, N, Ce and C for the component detection analysis of the synthesized parent and the doped material. Spectrometry (V-570, JASCO, Tokyo, Japan) was conducted to investigate the transmittance of the bare glass and thin film-coated glass substrate. The analysis range was from 250 to 800 nm, and analysis was performed using an integrating sphere.

### 2.3. Applied Anti-Counterfeiting and Fingerprinting

Anti-counterfeiting inks prepared using Ce^3+^-doped h-BN nanophosphors 1 wt% were dispersed in an aqueous solution containing 10 wt% of polyvinylpyrrolidone (PVP, M.W. = 14,000). The solution was spin coated on glass at 2000 rpm and banknotes were painted with a brush and then dried at 80 °C for an hour. The thin film coated on glass was photographed under UV light. Fingerprints were marked on glass substrates. Then Ce^3+^-doped h-BN nanophosphor powders were applied to the glass substrate surface and latent fingerprints on the surfaces were carefully wiped off. The latent fingerprints coated with the nanophosphors were developed using a UV lamp, and the appearance of the fingerprints was confirmed by photography.

## 3. Results and Discussion

### 3.1. Crystallinity and Morphology of Ce^3+^-Doped h-BN Nanophosphors

Ce^3+^-doped h-BN nanophosphors were synthesized by pyrolysis of a precursor in the form of a chemical compound prepared from boric acid, melamine and cerium nitrate at 900 °C under a nitrogen atmosphere. The samples were re-heated at 1100, 1200, 1300 and 1400 °C and their crystallinity was investigated by analysis of their XRD patterns as shown in Figure 2a. In all of the synthesized samples, the peak of the (002) phase matched the Joint Committee on Powder Diffraction Standards (JCPDS 34-0421) and the main diffraction peak of hexagonal boron nitride crystalline was observed in the XRD patterns. In the case of the sample synthesized at 900 °C, the shape of the (002) phase, which was the main diffraction peak, was slightly broad and showed a relatively weak signal. However, as the re-heating temperature increased, the XRD signal of the main diffraction peak (002) phase changed strongly and clearly. These changes were not the perfect form of h-BN for a sample synthesized at a relatively low temperature. Rather, the formed phase was turbo-stratic boron nitride (t-BN), which has the same crystal structure as h-BN but has hexagonal layers stacked and randomly rotated along the c-axis identified [19]. As the re-heating temperature was increased, the profiles of the diffraction peaks became clearer and their full width at half maximum (FWHM) narrowed (Figure 2b, black symbol); i.e., the crystallinity of t-BN formed by the pyrolysis of the sample synthesized at 900 °C improved as the re-heating temperature increased. The interplanar spacing of the (002) peak in the diffraction patterns of the Ce^3+^-doped h-BN nanophosphors gradually increased as the re-heating temperature increased. The FWHM of the samples decreased (Figure 2b, black symbol) and the interplanar spacing of the (002) phase increased (Figure 2b, red symbol).

Increased interplanar spacing and decreased FWHM of (002) peaks are commonly found during crystallization of h-BN [20]. We can see that the crystallinity of h-BN is improved with increased re-heating temperature. For these assumptions to be valid, Raman analysis expressed as the lattice frequency of an intrinsic constant was performed, and the results are shown in Figure 3. The Ce^3+^-doped h-BN nanophosphors’ position converts the E_2g_ mode near 1363 cm^−1^ of Raman spectra to low frequencies. Because the frequency of the Raman spectrum mode is inversely proportional to the square root of the constituent atomic mass [21], the transition to the lower frequency of the Raman spectrum means that heavy Ce atoms are incorporated into the BN lattice of light elements.

In addition, according to the re-heating temperature, the increased integrated area and decreased FWHM of the Raman shift indicate enhanced crystallinity, which is consistent with the XRD data. In the case of the specimen pyrolyzed at 900 °C and reheated at 1100 °C, Raman signals were hardly observed. However, when the reheating temperature was increased to 1200 °C or higher, the broad Raman signal strength became stronger and a distinct and strong Raman signal was observed at 1400 °C. This tendency is thought to be a result of the increase in crystallinity with increased reheating temperature.

Transmission electron microscopy was performed to observe the shape and morphology of the synthesized Ce^3+^-doped h-BN nanophosphors as shown in Figure 4. Figure 4a shows a sample pyrolyzed at 900 ℃ and the shape of the nanoparticles is unclear. When the lattice constant (0.332 nm) was profiled with high-resolution, it was found to be close to the XRD result. In addition, carbon was observed (Figure 4c) throughout the sample; this was unreacted and unburned material residue due to the relatively low heat-treatment temperature. However, the sample synthesized at 900 °C and re-heated at 1400 °C showed enhanced crystallinity according to the XRD results and showed a distinct elliptical plate shape of about 20-nm in the TEM image. The lattice constant d_(002)_ observed with high-resolution also decreased (0.334 nm) from the XRD data, and components of B, N and Ce were detected by EDX analysis (Figure 4d), confirming that Ce ions were doped in the h-BN lattice. The XRD and TEM results showed that, as the re-heating temperature increased, the enhanced crystallinity and the size of the particles increased [22]. It is thought to be an important process for synthesizing h-BN.

### 3.2. Luminescence of Ce^3+^-Doped h-BN Nanophosphors

Figure 5 shows the changes in luminescence according to the re-heating temperature of the Ce^3+^-doped h-BN synthesized at 900 °C. When the excitation wavelength was controlled at 304 nm, the photoluminescence (PL) wavelength showed deep-blue emission at 396 nm. The PL intensity was enhanced remarkably as the re-heating temperature increased as shown in Figure 5b. The increase in the PL intensity of the sample with increased re-heating temperature may be attributed to the substitutional incorporation of luminescent Ce^3+^ ions into the enhanced crystallinity lattice, as revealed in the previous section. The photoluminescence excitation (PLE) spectra of Ce^3+^-doped h-BN nanophosphors irradiated for emission at 396 nm showed an asymmetric curve centered at 304 nm. This is actually a mirror image of the PL spectra. This has been reported for organic dyes in the emission spectra [23]. Mirror symmetry is usually found for defect centers with weak Jahn-Teller interactions [24], which can be attributed to the substitution of Ce^3+^ ions for B atoms in the hexagonal boron nitride framework, as shown in Raman spectroscopy. The broad PL spectrum of cerium-doped nitride materials is usually attributed to the 4f-5d excitation of Ce^3+^ ions [25].

Because 5d electrons contribute to chemical bonds, the PL intensity and emission band are highly dependent on the chemical environment surrounding the Ce^3+^ ions [26]. As a result, the PL spectrum was highly dependent on the crystallinity of the host material. Interestingly, further increase in the re-heating temperature enhanced the crystallinity of Ce^3+^-doped h-BN nanophosphors.

### 3.3. Anti-Counterfeiting Application of Ce^3+^-Doped h-BN Nanophosphors

The large intrinsic bandgap (≥5 eV) of BN and the plate-like particle shape allow rare earth-doped boron nitride to be deposited as transparent phosphors on a flat surface. Figure 6a shows a Ce^3+^-doped h-BN nanophosphors thin film deposited on a glass substrate.

The transmittance of the glass substrate decreased slightly after coating with the phosphor thin film with a colloidal solution of Ce^3+^-doped h-BN nanophosphors. In daylight, it was difficult to distinguish between the bare glass and the phosphor-coated glass with the naked eye due to the transparency of the phosphor thin film. However, when irradiated with UV light, strong blue light emission can be seen from the phosphor film. The images of fingerprinting development are shown in the upper part of Figure 6b. The bare, donor and powdered images used Ce^3+^-doped h-BN nanophosphors re-heated at 1400 °C. The emission image was taken under UV light. The fingerprint was obtained from a person whose fingerprint has a whorl loop. The bare image was blurred in daylight. The powdered images produced by using prepared nanophosphors were eye-catching, because these particles were attached to the moisture component of the fingerprint. Under UV light, the blue emission confirmed that the contrast and resolution of the fingerprint had been improved. A Ce^3+^-doped h-BN nanophosphor colloidal solution was painted on a Korean bank note surface as shown in the lower images in Figure 6b. In daylight, it is difficult to distinguish between bare bank notes and those coated with the phosphor thin film. However, the phosphor-coated bank notes showed an intense blue emission text ‘Silla’ and label under UV light. Due to its light transmission and visible light emission under UV irradiation, this is hidden under normal conditions and can be recognized by the naked eye under UV radiation. Labels and text composed of h-BN based nanophosphors can be effectively hidden in daylight but can be easily identified under UV light, which is essential for a variety of anti-counterfeiting applications.

## 4. Conclusions

Hexagonal boron nitride doped with cerium ions was successfully synthesized by pyrolysis and re-heating of the precursor prepared from a chemical mixture of boric acid, melamine and cerium nitrate. Deep blue emission was detected at 396 nm under excitation with 304 nm from Ce^3+^-doped h-BN nanophosphors. The deep blue emission intensity was notably enhanced by increasing re-heating temperature. Increased re-heating temperature resulted in improved crystallinity and luminescence. It also affected particle growth. The deep blue emission is attributed to the transition of Ce^3+^ ions from the 5d level crystal field component to the 4f ground state. In the case of the specimen synthesized at 900 °C, the results of XRD patterns and Raman spectrum inferior crystallinity due to relatively low temperature were shown. However, as the reheating treatment temperature increased, the crystallinity improved, which is thought to be the effect of recrystallization caused by heat energy supplied from the outside. In addition, in the TEM image, particles with clear crystals were observed at 1400 °C, where the reheating treatment temperature was the highest and, as a result of component analysis, B, N and Ce were identified. This improvement in crystallinity has resulted in an increase in luminescence properties. The solution-based coating of Ce^3+^-doped h-BN nanophosphors was applied to various substrates; the films showed excellent transparency and luminescence with a strong visible blue color. The hiding and easy identification of nanophosphors under normal conditions demonstrates the feasibility of using rare earth-doped hexagonal boron nitride in anti-counterfeiting inks in a variety of applications.

## Figures and Tables

**Figure 1 materials-14-00193-f001:**
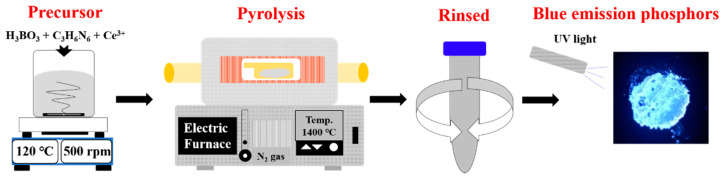
Schematic of preparation of the Ce^3+^-doped h-BN nanophosphors.

**Figure 2 materials-14-00193-f002:**
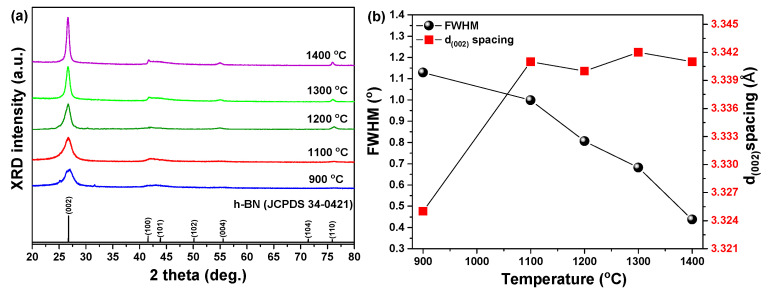
(**a**) X-ray powder diffraction (XRD) patterns, (**b**) full width at half maximum (FWHM) and interplanar spacing of (002) phase according to 900, 1100, 1200, 1300 and 1400 °C temperatures.

**Figure 3 materials-14-00193-f003:**
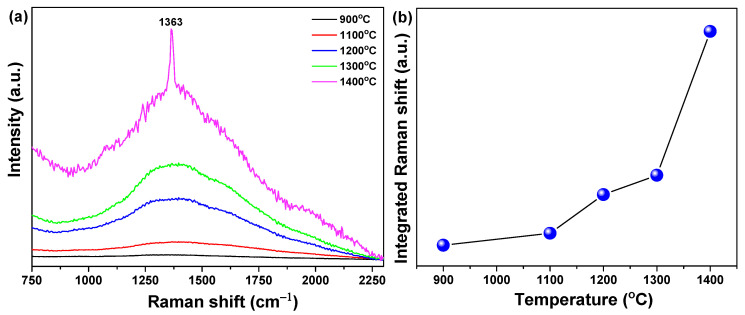
(**a**) Raman shift, (**b**) change of integrated and FWHM (inset) of Raman shift according to 900, 1100, 1200, 1300 and 1400 °C temperatures.

**Figure 4 materials-14-00193-f004:**
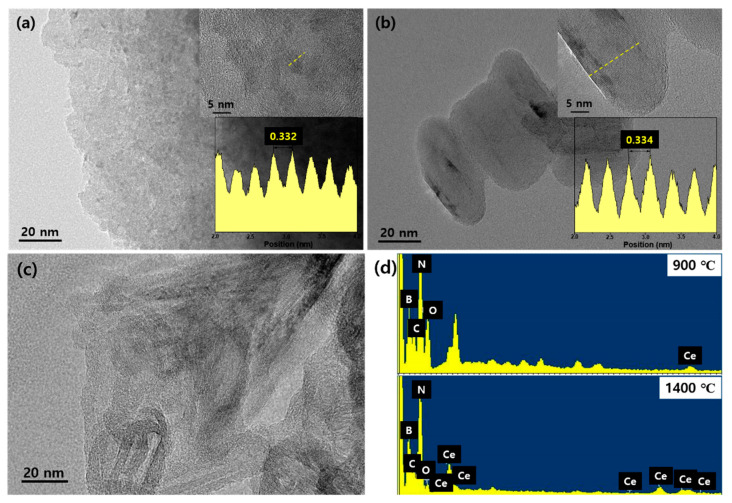
(**a**) Transmission electron microscopy (TEM) image of pyrolysis at 900 °C of Ce^3+^-dope h-BN nanophosphors (inset up; high-resolution, inset down; line profile), (**b**) TEM image of re-heating at 1400 °C of Ce^3+^-dope h-BN nanophosphors (inset up; high-resolution, inset down; line profile), (**c**) TEM image of observed remain carbon in pyrolysis at 900 °C and (**d**) X-ray spectroscopy (EDX) data of pyrolysis at different tempering temperatures (element identification: B, C, O, N and Ce).

**Figure 5 materials-14-00193-f005:**
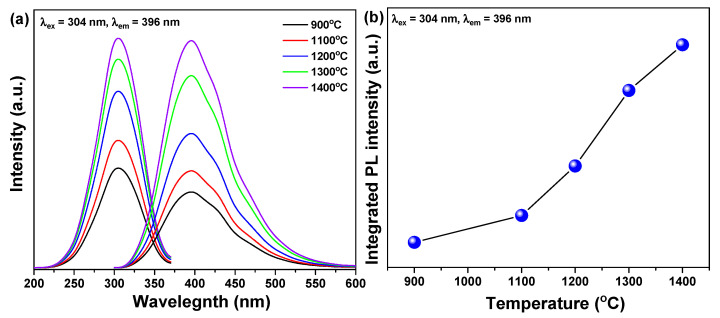
(**a**) Photoluminescence excitation (PLE) and photoluminescence (PL) spectra and (**b**) integrated PL intensity of Ce^3+^-doped h-BN nanophosphors in relation to re-heating temperatures of 900, 1100, 1200, 1300 and 1400 °C.

**Figure 6 materials-14-00193-f006:**
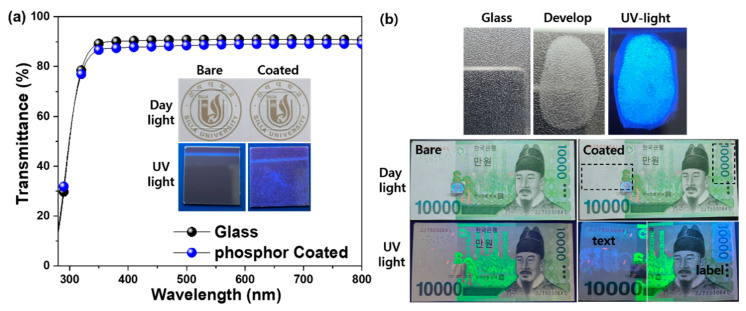
(**a**) Transmittance of Ce^3+^-doped h-BN nanophosphors coated on glass and (**b**) images of Ce^3+^-doped h-BN nanophosphors fingerprinting development on glass substrate (up) and painted on Korean bank notes (down); used samples were re-heated at 1400 °C.

## Data Availability

The data presented in this study are available in the database of the authors at the Faculty of Materials Science and Engineering.

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
