# Peer review of "Enhanced Crystallinity and Luminescence Characteristics of Hexagonal Boron Nitride Doped with Cerium Ions According to Tempering Temperatures"

_materials, 2021, doi:10.3390/ma14010193_

Round 1

Reviewer 1 Report

This is an interesting paper that combines: details of how re-heating improved crystallinity and luminescence peak power of hexagonal boron nitride doped  with cerium ions, with an application to prevent counterfeiting money by using a fluorescent ink made with the materials dissolved in it.

I would suggest to the authors to add a paragraph mentioning that an experimental study can be combined with a simulation effort to further understanding the materials using a relatively simple approach for luminescence such as found in

MF. Pereira, Analytical Expressions for Numerical Characterization of Semiconductors per Comparison with Luminescence, Materials Materials 2018, 11(1), 2; doi:10.3390/ma11010002  (2018)

I believe that this paper will be well received by the Materials community.

Author Response

Dear Reviewer,

We appreciate your kind consideration and the insightful comments on our recent submission (Manuscript ID: materials-1044022). Based on your opinion, we have corrected the manuscript and resubmit the revised version. We hope this revision can meet the reviewer’s requirements and accept the paper for publication. The followings are our point-by-point responses:

# Reviewer comments:

This is an interesting paper that combines: details of how re-heating improved crystallinity and luminescence peak power of hexagonal boron nitride doped with cerium ions, with an application to prevent counterfeiting money by using a fluorescent ink made with the materials dissolved in it. 

I would suggest to the authors to add a paragraph mentioning that an experimental study can be combined with a simulation effort to further understanding the materials using a relatively simple approach for luminescence such as found in 

  1. Pereira, Analytical Expressions for Numerical Characterization of Semiconductors per Comparison with Luminescence, Materials 2018, 11(1), 2; doi:10.3390/ma11010002 (2018) 

I believe that this paper will be well received by the Materials community.

# Response:

- In response to the comments, we have added the paper mentioned by the reviewer to the introductory part with reference No. 18.

All the comments pointed out by the reviewer have been revised and the revised text is marked in red for easy identification.

Very truly yours,

Donghyun Hwang

Reviewer 2 Report

Although the manuscript deals with an important topic, it is not prepared in accordance with the principles of writing scientific articles. in my opinion the paper is not acceptable for publication in its current form. To be able to consider it for publication, significant additions should be made, including among others:

- the study lacks a specific research goal, and there is also no indication of the validity of the research undertaken,

- the test methodology should be described so as to provide sufficient data to repeat the experiment under the same conditions,

- the methodology of sample preparation for TEM microscopy should be given,

- the parameters of the EDX analysis should be provided,

- EDX analysis spectra presented in Fig. 4d are unreadable, the designations of chemical elements are not visible, the specific chemical composition should be presented in more detail,

- conclusions are too general. It is necessary to compare the results obtained with the results from references,

- old references should be replaced by more recent journal paper references,

- language needs revision, language errors appear.

Author Response

Dear Reviewer,

We appreciate your kind consideration and the insightful comments on our recent submission (Manuscript ID: materials-1044022). Based on your opinion, we have corrected the manuscript and resubmit the revised version. We hope this revision can meet the reviewer’s requirements and accept the paper for publication. The followings are our point-by-point responses:

# Reviewer comments:

Although the manuscript deals with an important topic, it is not prepared in accordance with the principles of writing scientific articles. in my opinion the paper is not acceptable for publication in its current form. To be able to consider it for publication, significant additions should be made, including among others: 

  1. the study lacks a specific research goal, and there is also no indication of the validity of the research undertaken, 
  1. the test methodology should be described so as to provide sufficient data to repeat the experiment under the same conditions, 
  1. the methodology of sample preparation for TEM microscopy should be given, 
  1. the parameters of the EDX analysis should be provided, 
  1. EDX analysis spectra presented in Fig. 4d are unreadable, the designations of chemical elements are not visible, the specific chemical composition should be presented in more detail, 
  1. conclusions are too general. It is necessary to compare the results obtained with the results from references, 
  1. old references should be replaced by more recent journal paper references, 
  1. language needs revision, language errors appear.

#. Response to the first:

- In this study, the crystal structure change and luminescence properties of cerium ion doped hexagonal boron nitride at 900 ℃ were investigated at various reheat treatment temperatures. In addition, it is a thesis that suggests that boron hexagonal nitride, which has improved luminescence characteristics due to reheat treatment, can be applied to the field of counterfeiting. Please generously understand the parts that are not clearly stated. The introduction part has been modified.

#. Response to the second:

- The text was revised in the characteristic section by collecting the opinions of reviewers.

#. Response to the third:

- 0.01 g of the synthesized specimen was added to a beaker containing 1 ml of ethyl alcohol, and then dispersed for about 30 minutes using an ultrasonicator. The copper grid was held with a tweezer, slightly immersed in the solution, and dried for about 1 minute on a hot plate prepared at a temperature of 80 °C. Thereafter, surface impurities were removed using a nitrogen gun.

#. Response to the fourth:

- The electrons generated by the application of current are accelerated and collide with the sample. At this time, inside the sample, internal electrons bounce to the outside by incident electrons, and the high magnification image of the sample is observed using the bounced secondary electrons. At this time, in order to lower the thermodynamic energy of the atom, the electrons in the upper orbit are transferred to the vacant position to stabilize the atom. By measuring this energy, the qualitative component analysis of the sample is performed. The resolution was about 129 eV and the analysis time was X-ray exposure for 2 minutes for each specimen. At this time, the components were compared with B, N, Ce, and C for the component detection analysis of the synthesized parent and the doped material.

#. Response to the fifth:

- The figure was revised by collecting the opinions of the reviewers, and the contents related to the text were described. As a result of EDX analysis, it can be confirmed that the specimen synthesized at 900 °C was detected more than the component of Ce doped by the remaining carbon. Observed as in 4c. However, in the case of the specimen reheated at 1400 °C, it was found that B, N, and Ce were clearly observed as a result of the EDX component analysis.

#. Response to the sixth:

- The reviewer's opinions were collected, and the conclusions were attached. ‘In the case of the specimen synthesized at 900 °C, the results of XRD patterns and Raman spectrum inferior crystallinity due to relatively low temperature were shown. However, as the reheat treatment temperature increased, the crystallinity improved, which is thought to be the effect of recrystallization caused by heat energy supplied from the outside. In addition, in the TEM image, particles with clear crystals were observed at 1400 °C, where the reheat treatment temperature was the highest, and as a result of component analysis, B, N, and Ce were identified. This improvement in crystallinity has resulted in an increase in luminescence properties.’

#. Response to the seventh:

- Revision was completed by collecting reviewers' opinions. Line 342.

#. Response to the Eigth:

- This is the text submitted after completing proofreading. Thank you for your understanding.

All the comments pointed out by the reviewer have been revised and the revised text is marked in red for easy identification.

Very truly yours,

Donghyun Hwang

Reviewer 3 Report

Dear Authors,
Thank you for sending your paper for publication. I found your paper very interesting and of high scientific quality and a high level of presentation. I found only a few small flaws. I think that:
in line 72 you should remove one "oven"
in line 215 you should remove one "blue"
and finally, I cannot find a reference to position number 26 so remove it from the list or refer to it in the text.

Author Response

Dear Reviewer,

We appreciate your kind consideration and the insightful comments on our recent submission (Manuscript ID: materials-1044022). Based on your opinion, we have corrected the manuscript and resubmit the revised version. We hope this revision can meet the reviewer’s requirements and accept the paper for publication. The followings are our point-by-point responses:

# Reviewer comments:

Thank you for sending your paper for publication. I found your paper very interesting and of high scientific quality and a high level of presentation. I found only a few small flaws. I think that: 

  1. in line 72 you should remove one "oven" 
  1. in line 215 you should remove one "blue" 
  1. and finally, I cannot find a reference to position number 26 so remove it from the list or refer to it in the text.

#. Response to the first:

- Modified by referring to reviewer's opinion.

#. Response to the second:

- Modified by referring to reviewer's opinion

#. Response to the third:

- Modified by referring to reviewer's opinion

All the comments pointed out by the reviewer have been revised and the revised text is marked in red for easy identification.

Very truly yours,

Donghyun Hwang

Reviewer 4 Report

The work by Jung et al. investigated enhanced crystallinity and luminescence characteristics of hexagonal boron nitride doped with cerium ions by tempering temperatures. The work is very useful. However, there remain some issues. I can’t recommend the acceptance for the current form unless all my concerns below are well responded:

  1. Motivation is not clear. Why you are going to use re-heating, what is the principle embeded, especially, the synthesis has firstly reported in ref. 17? It is meaningless for your emphasis of the synthesis in the introduction.
  2. The description of the experiments is not that clear. For the rinsing and centrifuge, the solvent remains DI Water? For the tempering process, how did you perform? How long/fast will it take?
  3. The quality of Fig. 1 is pretty bad, especially many words are not clear. Please improve it.
  4. The author claimed that Increased interplanar spacing and decreased FWHM of (002) peaks are commonly found, but they do not give out the exact reason for it. It is fully the same as reference or some minor change?
  5. In Fig. 3, only for the sample tempered at 1400oC, there is narrow and sharp peak at 1363cm-1. It is abnormal. So why does it only happen for 1400oC? The result indicates that others are all increased, but do not have sharp peak. This peak is really matched the materials characteristic peak, as the author cited the basic principle for the change.
  6. In Fig. 4, it is not clear what the periodic lattice belongs as all the materials are the same. But only some areas can see such evident periodic structure.
  7. The English is casual at some places. The authors check and improve it carefully and thoroughly.

Author Response

Dear Reviewer,

We appreciate your kind consideration and the insightful comments on our recent submission (Manuscript ID: materials-1044022). Based on your opinion, we have corrected the manuscript and resubmit the revised version. We hope this revision can meet the reviewer’s requirements and accept the paper for publication. The followings are our point-by-point responses:

# Reviewer comments:

The work by Jung et al. investigated enhanced crystallinity and luminescence characteristics of hexagonal boron nitride doped with cerium ions by tempering temperatures. The work is very useful. However, there remain some issues. I can’t recommend the acceptance for the current form unless all my concerns below are well responded: 

  1. Motivation is not clear. Why you are going to use re-heating, what is the principle embeded, especially, the synthesis has firstly reported in ref. 17? It is meaningless for your emphasis of the synthesis in the introduction. 
  1. The description of the experiments is not that clear. For the rinsing and centrifuge, the solvent remains DI Water? For the tempering process, how did you perform? How long/fast will it take? 
  1. The quality of Fig. 1 is pretty bad, especially many words are not clear. Please improve it. 
  1. The author claimed that Increased interplanar spacing and decreased FWHM of (002) peaks are commonly found, but they do not give out the exact reason for it. It is fully the same as reference or some minor change? 
  1. In Fig. 3, only for the sample tempered at 1400 oC, there is narrow and sharp peak at 1363 cm-1. It is abnormal. So why does it only happen for 1400 oC? The result indicates that others are all increased, but do not have sharp peak. This peak is really matched the materials characteristic peak, as the author cited the basic principle for the change. 
  1. In Fig. 4, it is not clear what the periodic lattice belongs as all the materials are the same. But only some areas can see such evident periodic structure. 
  1. The English is casual at some places. The authors check and improve it carefully and thoroughly. 

#. Response to the first:

- In this study, the crystal structure change and luminescence properties of cerium ion doped hexagonal boron nitride at 900 ℃ were investigated at various reheat treatment temperatures. In addition, it is a thesis that suggests that boron hexagonal nitride, which has improved luminescence characteristics due to reheat treatment, can be applied to the field of counterfeiting. Please generously understand the parts that are not clearly stated. The introduction part has been modified.

#. Response to the second:

- After heat treatment, rinsed with D.I water was performed to remove unreacted substances. When the powder was collected by agglomeration in the direction of gravity using a centrifuge, the solution was discarded and the recovered powder specimen was dried at 80 °C for 24 hours. In addition, the reheating temperature was conducted at 1100, 1200, 1300, 1400 ℃ for 2 hours each in a nitrogen atmosphere.

The manuscript has been modified. Line 77.

#. Response to the third:

- The figure was modified by collecting the opinions of reviewers.

#. Response to the fourth:

- It is generally known that the interplanar spacing increases when rare earth ions with a large ionic radius are doped in the lattice, but in this study, the re-arrangement of the lattice due to external thermal energy by reheating rather than the change of the lattice constant due to rare earth ions. I think the influence is greater. As a result of this, I think that the FWHM of the (002) peak decreases and the signal strength increases, and it is not completely the same as the reference. I will try to do more research to find out the clear facts.

#. Response to the fifth:

- We know that the Raman signal is expressed as the natural frequency of the grid. These results are related to the XRD results, and it is believed that the crystallinity that increases as the reheat treatment temperature increases is expressed as the natural frequency of BN. Therefore, at relatively low temperatures, broad grating vibrations appear, and the reheated specimen at 1400 °C has the best crystallinity, so it is thought that a clear grating frequency peak was found.

#. Response to the sixth:

- TEM images show specimens heat-treated at 900 degrees Celsius and 1400 degrees Celsius, respectively. In the case of 900 degrees Celsius, the shape of carbon was observed throughout the sampled specimen, leaving unreacted material due to relatively low temperature, and also the clear shape and lattice of the particles could not be observed. However, in the case of the specimen reheated at 1400°C, no unreacted material was observed due to the relatively high temperature, and the clear shape and lattice of the particles could be observed. If we consider these TEM shape observations and lattice observations in relation to the XRD results, we think that the reheat treatment temperature improves the crystallinity and makes it possible to observe the distinct lattice and particle shape.

#. Response to the seventh:

- This is the text submitted after completing proofreading. Thank you for your understanding.

All the comments pointed out by the reviewer have been revised and the revised text is marked in red for easy identification.

Very truly yours,

Donghyun Hwang

Round 2

Reviewer 2 Report

After introducing changes by the authors and providing satisfactory answers, in my opinion the article is acceptable for publication.